# MEMO: Test Time Robustness via Adaptation and Augmentation

**Marvin Zhang**[1], **Sergey Levine**[1], **Chelsea Finn**[2]
[1]UC Berkeley [2]Stanford University

## Abstract

While deep neural networks can attain good accuracy on in-distribution test points, many applications require robustness even in the face of unexpected perturbations in the input, changes in the domain, or other sources of distribution shift. We study the problem of *test time robustification*, i.e., using the test input to improve model robustness. Recent prior works have proposed methods for test time adaptation, however, they each introduce additional assumptions, such as access to multiple test points, that prevent widespread adoption. In this work, we aim to study and devise methods that make no assumptions about the model training process and are broadly applicable at test time. We propose a simple approach that can be used in any test setting where the model is probabilistic and adaptable: when presented with a test example, perform different data augmentations on the data point, and then adapt (all of) the model parameters by minimizing the entropy of the model's average, or *marginal*, output distribution across the augmentations. Intuitively, this objective encourages the model to make the same prediction across different augmentations, thus enforcing the invariances encoded in these augmentations, while also maintaining confidence in its predictions. In our experiments, we evaluate two baseline ResNet models, two robust ResNet-50 models, and a robust vision transformer model, and we demonstrate that this approach achieves accuracy gains of 1-8% over standard model evaluation and also generally outperforms prior augmentation and adaptation strategies. For the setting in which only one test point is available, we achieve state-of-the-art results on the ImageNet-C, ImageNet-R, and, among ResNet-50 models, ImageNet-A distribution shift benchmarks.

## 1 Introduction

Deep neural network models have achieved excellent performance on many machine learning problems, such as image classification, but are often brittle and susceptible to issues stemming from *distribution shift*. For example, deep image classifiers may degrade precipitously in accuracy when encountering input perturbations, such as noise or changes in lighting [12] or domain shifts which occur naturally in real world applications [21]. Therefore, robustification of deep models against these test shifts is an important and active area of study.

Most prior works in this area have focused on techniques for training time robustification, including utilizing larger models and datasets [34], various forms of adversarial training [39, 48], and aggressive data augmentation [51, 13, 24, 14]. Employing these techniques requires modifying the training process, which may not be feasible if, e.g., it involves heavy computation or non public data. Furthermore, these techniques do not rely on any information about the test points that the model must predict on, even though these test points may provide significant information for improving model robustness. Recently, several works have proposed methods for improving accuracy via *adaptation* after seeing the test data, typically by updating a subset of the model's weights [44, 45, 18], normalization statistics [40], or both [46, 52]. Though effective at handling test shifts, these

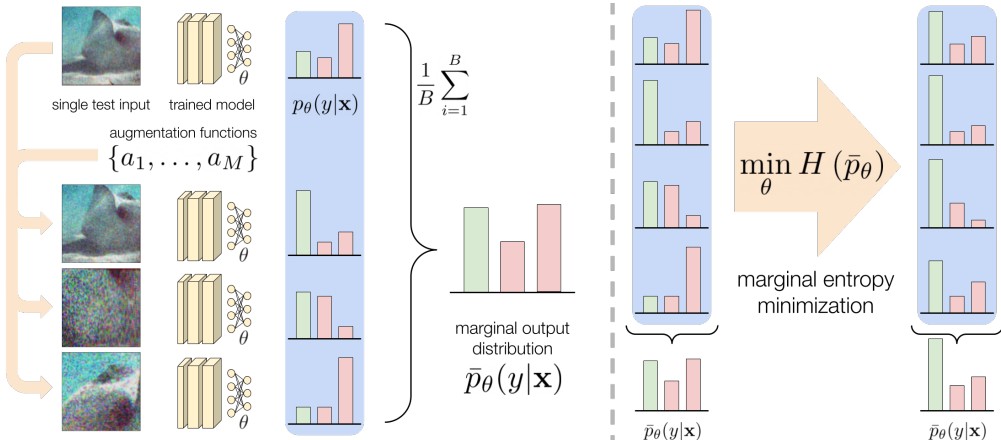

Figure 1: Left: at test time, as detailed in Section 3, we have a single test input $\mathbf{x}$, a set of data augmentation functions $\{a_1, \ldots, a_M\}$, and a trained model that outputs a probabilistic predictive distribution and has adaptable parameters $\theta$. We perform different augmentations on $\mathbf{x}$ and pass these augmented inputs to the model in order to estimate the marginal output distribution averaged over augmentations. Right: we perform a gradient update on the model to minimize the entropy of this marginal distribution, thus encouraging the model predictions to be invariant across different augmentations while maintaining confident predictions. The final prediction is then made on the original data point, i.e., the predictive distribution in the top right of the schematic.

methods sometimes still require specialized training procedures, and they typically rely on extracting information via batches or even entire sets of test inputs, thus introducing additional assumptions.

In this work, we focus on methods for *test time robustness*, in which the specific test input may be leveraged in order to improve the model's prediction on that point. We are interested in studying and devising methods for improving model robustness that are "plug and play", i.e., they can be readily used with a wide variety of pretrained models and test settings. We also want methods that synergize with other robustification techniques, in order to achieve greater performance than using either in isolation. With these goals in mind, we devise a novel test time robustness method based on adaptation and augmentation. As illustrated in Figure 1, when presented with a test point, we adapt the model by augmenting the test point in different ways while encouraging the model to make consistent predictions, thus respecting the invariances encoded in the data augmentations. We further encourage the model to make confident predictions, thus arriving at the proposed method: minimize the *marginal entropy* of the model's predictions across the augmented versions of the test point.

We refer to the proposed method as **m**arginal **e**ntropy **m**inimization with **o**ne test point (MEMO), and this is the primary contribution of our work. MEMO makes direct use of pretrained models without any assumptions about their particular training procedure or architecture, while requiring only a single test input for adaptation. In Section 4, we demonstrate empirically that MEMO consistently improves the performance of ResNet [11] and vision transformer [7] models on several challenging ImageNet distribution shift benchmarks, achieving several new state-of-the-art results for these models in the setting in which only one test point is available. MEMO consistently outperforms non adaptive marginal distribution predictions (between 1-10% improvement) on the ImageNet-C [12] and ImageNet-R [14] test sets, indicating that adaptation plays a crucial role in improving predictive accuracy. MEMO encourages both invariance across augmentations and confident predictions, and an ablation study in Section 4 shows that both components are important for maximal performance gains. Also, MEMO is, to the best of our knowledge, the first adaptation method to improve performance (by 1-4% over standard model evaluation) on the ImageNet-A test set [15].

## 2  Related work

Distribution shift has been studied under a number of frameworks [36], including domain adaptation [41, 6, 47], domain generalization [5, 32, 9], and distributionally robust optimization [4, 16, 39]. These frameworks typically leverage additional training or test assumptions in order to make the distribution shift problem more tractable. Largely separate from these frameworks, various empirical methods have also been proposed for dealing with shift, such as increasing the model and training dataset

size or using heavy training augmentations [34, 51, 14]. The focus of this work is complementary to these efforts: MEMO is applicable to a wide range of pretrained models, including those trained via robustness methods, and can achieve further performance gains via test time adaptation.

Prior test time adaptation methods generally either make significant training or test time assumptions. Some methods update the model using batches or even entire datasets of test inputs, such as by computing batch normalization (BN) statistics on the test set [25, 19, 33, 40], computing class prototypes [18], or minimizing the (conditional) entropy of model predictions across a batch of test data [46]. The latter approach is closely related to MEMO. The differences are that MEMO minimizes *marginal* entropy using single test points and data augmentation and adapts all of the model parameters rather than just those associated with normalization layers, thus not requiring multiple test points or specific model architectures. Other test time adaptation methods can be applied to single test points but require specific training procedures or models [44, 17, 40, 1, 3]. Test time training (TTT) [44] requires a specialized model with a rotation prediction head and a different procedure for training this model. Schneider et al. [40] show that BN adaptation can be effective even with only one test point. As we discuss in Section 3, MEMO synergizes well with this technique of "single point" BN adaptation. Mao et al. [29] propose a test time adaptation method based on input perturbations for robustness to adversarial attacks. Concurrently with our work, Sivaprasad and Fleuret [43] propose a similar method for test time adaptation by encouraging invariance to data augmentations, and they test their method on the corrupted CIFAR and VisDA [35] datasets.

A number of works have noted that varying forms of strong data augmentation on the training set can improve the resulting model's robustness [51, 13, 24, 14]. Data augmentations are also sometimes used on the test data directly by averaging the model's outputs across augmented copies of the test point [23, 42], i.e., predicting according to the model's marginal output distribution. When using cropping as the augmentation, this technique is often referred to as multicrop evaluation [22]. We instead use the term test time augmentation (TTA), as we use additional augmentations beyond cropping [13]. TTA has been shown to be useful both for improving model accuracy and calibration [2] as well as handling distribution shift [31]. We take this idea one step further by explicitly adapting the model such that its marginal output distribution has low entropy. This extracts an additional learning signal for improving the model, and furthermore, the adapted model can then make its final prediction on the clean test point rather than the augmented copies. We empirically show in Section 4 that these differences lead to improved performance over this non adaptive TTA baseline.

## 3   Augmenting and Adapting at Test Time

Data augmentations are typically used to train the model to respect certain invariances – e.g., changes in lighting or viewpoint do not change the underlying class label – but, especially when faced with distribution shift, the model is not guaranteed to obey the same invariances at test time. In this section, we introduce MEMO, a method for test time robustness that adapts the model such that it respects these invariances on the test input. We use "test time robustness" specifically to refer to techniques that operate directly on pretrained models and single test inputs – single point BN adaptation and TTA, as described in Section 2, are examples of prior test time robustness methods.

In the test time robustness setting, we are given a trained model $f_\theta$ with parameters $\theta \in \Theta$. We do not require any special training procedure and do not make any assumptions about the model, except that $\theta$ is adaptable and that $f_\theta$ produces a conditional output distribution $p_\theta(y|\mathbf{x})$ that is differentiable with respect to $\theta$.[1] All standard deep neural network models satisfy these assumptions. A single point $\mathbf{x} \in \mathcal{X}$ is presented to $f_\theta$, for which it must predict a label $\hat{y} \in \mathcal{Y}$ immediately. Note that this is precisely identical to the standard test time inference procedure for regular supervised learning models – in effect, we are simply modifying how inference is done, without any additional assumptions on the training process or on test time data availability. This makes test time robustness methods a simple "slot-in" replacement for the ubiquitous and standard test time inference process. We assume sampling access to a set of augmentation functions $\mathcal{A} \triangleq \{a_1, \dots, a_M\}$ that can be applied to the test point $\mathbf{x}$. We use these augmentations and the self-supervised objective detailed below to adapt the model before it predicts on $\mathbf{x}$. When given a set of test inputs, the model adapts and predicts on each test point independently. We do not assume access to any ground truth labels.

---

[1]Single point BN adaptation also assumes that the model has BN layers – as shown empirically in Section 4, this is an assumption that we do not require but can also benefit from.

---

**Algorithm 1** Test time robustness via MEMO

---

**Require:** trained model $f_\theta$, test point $\mathbf{x}$, number of augmentations $B$, learning rate $\eta$, update rule $G$
 1: Sample $a_1, \ldots, a_B \overset{\text{i.i.d.}}{\sim} \mathcal{U}(\mathcal{A})$ and produce augmented points $\tilde{\mathbf{x}}_i = a_i(\mathbf{x})$ for $i \in \{1, \ldots, B\}$
 2: Compute estimate $\tilde{p} = \frac{1}{B} \sum_{i=1}^{B} p_\theta(y|\tilde{\mathbf{x}}_i) \approx \bar{p}_\theta(y|\mathbf{x})$ and $\tilde{\ell} = H(\tilde{p}) \approx \ell(\theta; \mathbf{x})$, i.e., Eq. 2
 3: Adapt parameters via update rule $\theta' \leftarrow G(\theta, \eta, \tilde{\ell})$
 4: Predict $\hat{y} \triangleq \arg\max_y p_{\theta'}(y|\mathbf{x})$

---

### 3.1 Marginal Entropy Minimization with One test point

Given a test point $\mathbf{x}$ and set of augmentation functions $\mathcal{A}$, we sample $B$ augmentations from $\mathcal{A}$ and apply them to $\mathbf{x}$ in order to produce a batch of augmented data $\tilde{\mathbf{x}}_1, \ldots, \tilde{\mathbf{x}}_B$. The model's average, or *marginal*, output distribution with respect to the augmented points is given by

$$\bar{p}_\theta(y|\mathbf{x}) \triangleq \mathbb{E}_{\mathcal{U}(\mathcal{A})}\left[p_\theta(y|a(\mathbf{x}))\right] \approx \frac{1}{B} \sum_{i=1}^{B} p_\theta(y|\tilde{\mathbf{x}}_i), \tag{1}$$

where the expectation is with respect to uniformly sampled augmentations $a \sim \mathcal{U}(\mathcal{A})$.

What properties do we desire from this marginal distribution? To answer this question, consider the role that data augmentation typically serves during training. For each training point $(\mathbf{x}^{\text{train}}, y^{\text{train}})$, the model $f_\theta$ is trained using multiple augmented forms of the input $\tilde{\mathbf{x}}_1^{\text{train}}, \ldots, \tilde{\mathbf{x}}_E^{\text{train}}$. $f$ is trained to obey the invariances between the augmentations and the label – no matter the augmentation on $\mathbf{x}^{\text{train}}$, $f$ should predict, with confidence, the same label $y^{\text{train}}$. We seek to devise a similar learning signal during test time, without any ground truth labels. That is, after adapting:

(1) the model $f_\theta$ predictions should be invariant across augmented versions of the test point, and
(2) the model $f_\theta$ should be confident in its predictions, even for heavily augmented versions of the test point, since all versions have the same underlying label.

Optimizing the model for more confident predictions can be justified from the assumption that the true underlying decision boundaries between classes lie in low density regions of the data space [8]. With these two goals in mind, we propose to adapt the model using the entropy of its marginal output distribution over augmentations (Eq. 1), i.e.,

$$\ell(\theta; \mathbf{x}) \triangleq H\left(\bar{p}_\theta(\cdot|\mathbf{x})\right) = -\sum_{y \in \mathcal{Y}} \bar{p}_\theta(y|\mathbf{x}) \log \bar{p}_\theta(y|\mathbf{x}). \tag{2}$$

Note that this objective is not the same as optimizing the average conditional entropy of the model's predictive distributions across augmentations, i.e.,

$$\ell_{\text{CE}}(\theta; \mathbf{x}) \triangleq \frac{1}{B} \sum_{i=1}^{B} H(p_\theta(\cdot|\tilde{\mathbf{x}}_i)). \tag{3}$$

A model which predicts confidently but *differently* across augmentations would minimize Eq. 3 but not Eq. 2. Optimizing Eq. 2 encourages both confidence and invariance, since the entropy of $\bar{p}_\theta(\cdot|\mathbf{x})$ is minimized when the model outputs the same (confident) prediction regardless of the augmentation.

Algorithm 1 presents the overall method MEMO for test time adaptation. Though prior test time adaptation methods must carefully choose which parameters to adapt in order to avoid degenerate solutions [46], our adaptation procedure simply adapts all of the model's parameters $\theta$ (line 3). Given that $p_\theta(y|\mathbf{x})$ is differentiable with respect to $\theta$, we can directly use gradient based optimization to adapt $\theta$ according to Eq. 2. We use only one gradient step per test point, because empirically we found this to be sufficient for improved performance while being more computationally efficient. After this step, we use the adapted model $f_{\theta'}$ to predict on the original test input $\mathbf{x}$ (line 4).

### 3.2 Composing MEMO with Prior Methods

An additional benefit of MEMO is that it synergizes with other approaches for handling distribution shift. In particular, MEMO can be composed with prior methods for training robust models and adapting model statistics, thus leveraging the performance improvements of each technique.

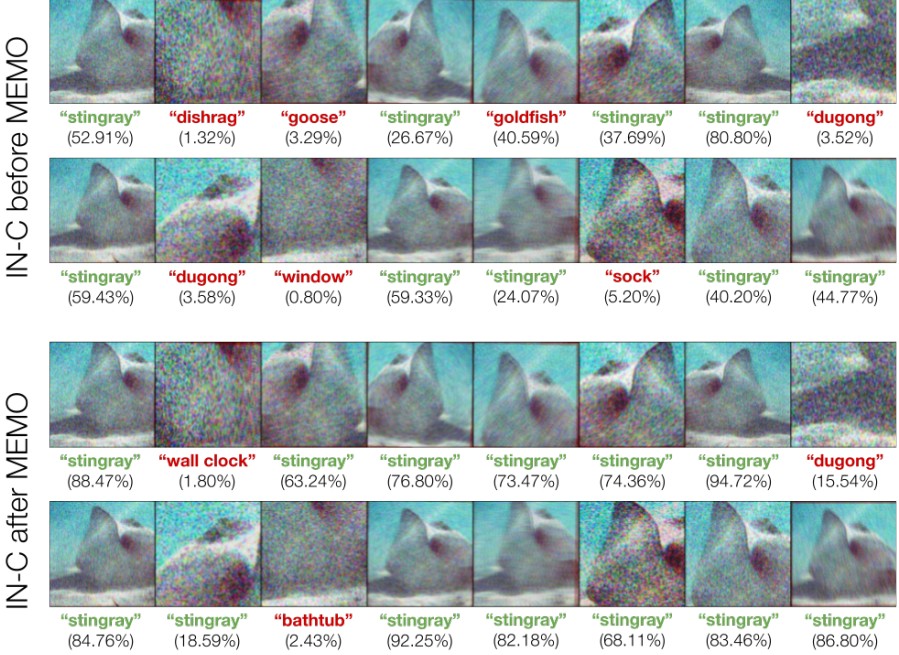

Figure 2: We visualize augmentations of a randomly chosen data point from the "Gaussian Noise level 3" ImageNet-C test set. Even for a robust model trained with heavy data augmentations [14], both its predictive accuracy and confidence (as shown in the top two rows) drop sharply when encountering test shift. As shown in the bottom two rows, these drops can be remedied via MEMO adaptation.

**Pretrained robust models.** Since MEMO makes no assumptions about, or modifications to, the model training procedure, performing adaptation on top of pretrained robust models, such as those trained with heavy data augmentations, is as simple as using any other pretrained model. Crucially, we find that, in practice, the set of test augmentations $\mathcal{A}$ does not have to match the augmentations that were used to train the model. For simplicity and efficiency, we use augmentations that can be easily sampled and are applied directly to the model input $\mathbf{x}$. These properties do not hold for, e.g., data augmentation techniques based on image translation models, such as DeepAugment [14], or feature mixing, such as moment exchange [24]. However, we can still use models trained with these data augmentation techniques as our starting point for adaptation, thus allowing us to improve upon their state-of-the-art results. As noted above, using pretrained models is not as easily accomplished for adaptation methods which require complicated or specialized training procedures and model architectures, such as TTT [44] or ARM [52]. In our experiments, we use AugMix as our set of augmentations [13], as it satisfies the above properties and still yields significant diversity when applied, as depicted in Figure 2. Note that AugMix explicitly does not use augmentations that are similar to the corruptions in the CIFAR-10-C and ImageNet-C test sets [12, 13].

**Adapting BN statistics.** Schneider et al. [40] showed that, even when presented with just a single test point, partially adapting the estimated mean and variance of the activations in each batch normalization (BN) layer of the model can still be effective in some cases for handling distribution shift. In this setting, to prevent overfitting to the test point, the channelwise mean and variance $[\boldsymbol{\mu}_{\text{test}}, \boldsymbol{\sigma}^2_{\text{test}}]$ estimated from this point are mixed with the the mean and variance $[\boldsymbol{\mu}_{\text{train}}, \boldsymbol{\sigma}^2_{\text{train}}]$ computed during training according to a prior strength $N$. That is, for $\boldsymbol{\nu} \in \{\boldsymbol{\mu}, \boldsymbol{\sigma}^2\}$,

$$\boldsymbol{\nu} \triangleq \frac{N}{N+1}\boldsymbol{\nu}_{\text{train}} + \frac{1}{N+1}\boldsymbol{\nu}_{\text{test}}.$$

This technique is also straightforward to combine with MEMO: we simply use the adapted BN statistics whenever computing the model's output distribution. We find in our experiments that this technique never degrades, and generally improves, the performance of test time adaptation, thus we combine MEMO with this technique by default whenever applicable. Following the suggestion in Schneider et al. [40], we set $N = 16$ for all of our experiments in the next section.

# 4 Experiments

Our experiments aim to answer the following questions:

(1) How does MEMO compare to prior methods for test time adaptation and test time robustness?
(2) Can MEMO be combined with a wide range of model architectures and pretraining methods?
(3) Which aspect of MEMO, the adaptation or augmentation, is the most important?

We evaluate MEMO on a total of five distribution shift benchmarks. We conduct CIFAR-10 [22] experiments on the CIFAR-10-C [12] and CIFAR-10.1 [37] test sets, and we conduct ImageNet [38] experiments on the ImageNet-C [12], ImageNet-R [14], and ImageNet-A [15] test sets.

To answer question (1), we compare to test time training (TTT) [44] in the CIFAR-10 experiments, for which we train ResNet-26 models following their protocol and specialized architecture. We do not compare to TTT for the ImageNet experiments due to the computational demands of training state-of-the-art models and because Sun et al. [44] do not report competitive ImageNet results. For the ImageNet experiments, we compare to Tent [46] and BN adaptation, which can be used with pretrained models but require multiple test inputs (or even the entire test set) for adaptation. We provide BN adaptation with $256$ test inputs at a time and set the prior strength $N = 256$ [40].

For Tent, we use test batch sizes of $64$ and, for ResNet-50 models, test both "online" adaptation – where the model adapts continually through the entire evaluation – and "episodic" adaptation – where the model is reset after each test batch [46]. Note that the evaluation protocols are different for these two methods: whereas MEMO is tasked with predicting on each test point immediately after adaptation, BN adaptation predicts on a batch of 256 test points after computing BN statistics on the batch, and Tent predicts on a batch of 64 inputs after adaptation but also, in the online setting, continually adapts throughout evaluation. In all experiments, we further compare to single point BN adaptation [40] and the TTA baseline that simply predicts according to $\bar{p}_\theta(y|\mathbf{x})$ (Eq. 1) [23, 2]. Full details on our experimental protocol are provided in Appendix A.

To answer question (2), we apply MEMO on top of multiple pretrained models with different architectures, trained via several different procedures. For CIFAR-10, we train our own ResNet-26 [11] models. For ImageNet, we use the best performing ResNet-50 robust models from prior work, which includes those trained with DeepAugment and AugMix augmentations [14] as well as those trained with moment exchange and CutMix [24]. To evaluate the generality of prior test time robustness methods and MEMO, we also evaluate the small robust vision transformer (RVT*-small), which provides superior performance on all three ImageNet distribution shift benchmarks compared to the robust ResNet-50 models [30]. Finally, we evaluate ResNext-101 models [50, 28] on ImageNet-A, as these models previously achieved the strongest results for this test set [14].

Finally, to answer (3), we conduct ablative studies in subsection 4.2: first to determine the relative importance of maximizing confidence (via entropy minimization) versus enforcing invariant predictions across augmented copies of each test point, second to determine the importance of the particular augmentation functions used, and third to determine the required number of augmented samples per inference. The comparison to the non adaptive TTA baseline also helps determine whether simply augmenting the test point is sufficient or if adaptation is additionally helpful. In Appendix B, we provide further experiments ablating the augmentation component specifically.

## 4.1 Main Results

We summarize results for CIFAR-10, CIFAR-10.1, and CIFAR-10-C in Table 1, with full CIFAR-10-C results in Appendix C. We use indentations to indicate composition, e.g., TTT is performed at test time on top of their specialized joint training procedure. Across all corruption types in CIFAR-10-C, MEMO consistently improves test error compared to the baselines, non adaptive TTA, and TTT. MEMO also provides a larger performance gain on CIFAR-10.1 compared to TTT. We find that the non adaptive TTA baseline is competitive for these relatively simple test sets, though it is worse than MEMO for CIFAR-10-C. Of these three test sets, CIFAR-10-C is the only benchmark that explicitly introduces distribution shift, which suggests that adaptation is useful when the test shifts are more prominent. Both TTA and MEMO are also effective at improving performance for the original CIFAR-10 test set where there is no distribution shift, providing further support for the widespread use of augmentations in standard evaluation protocols [23, 2].

Table 1: Results for CIFAR-10, CIFAR-10.1, and CIFAR-10-C. *Results from Sun et al. [44].

| | CIFAR-10 Error (%) | CIFAR-10.1 Error (%) | CIFAR-10-C Average Error (%) |
|---|---|---|---|
| ResNet-26 [11] | 9.2 | 18.4 | 22.5 |
| + TTA | **7.3** (−1.9) | 14.8 (−3.6) | 19.9 (−2.6) |
| + MEMO (ours) | **7.3** (−1.9) | **14.7** (−3.7) | **19.6** (−2.9) |
| + Joint training* [44] | 8.1 | 16.7 | 22.8 |
| + TTT* [44] | 7.9 (−0.2) | 15.9 (−0.8) | 21.5 (−1.3) |

Table 2: Test results for the ImageNet test sets. MEMO achieves new state-of-the-art performance on each benchmark for ResNet-50 models for the single test point setting. For RVT*-small, MEMO improves performance across all benchmarks and reaches a new state of the art for ImageNet-C and ImageNet-R. Compared to prior approaches, MEMO offers more consistent improvements.

| | ImageNet-C mCE ↓ | ImageNet-R Error (%) | ImageNet-A Error (%) |
|---|---|---|---|
| Baseline ResNet-50 [11] | 76.7 | 63.9 | 100.0 |
| + TTA | 77.9 (+1.2) | 61.3 (−2.6) | 98.4 (−1.6) |
| + Single point BN | 71.4 (−5.3) | 61.1 (−2.8) | 99.4 (−0.6) |
| + MEMO (ours) | 69.9 (−6.8) | 58.8 (−5.1) | 99.1 (−0.9) |
| + BN ($N = 256, n = 256$) | 61.6 (−15.1) | 59.7 (−4.2) | 99.8 (−0.2) |
| + Tent (online) [46] | 54.4 (−22.3) | 57.7 (−6.2) | 99.8 (−0.2) |
| + Tent (episodic) | 64.7 (−12.0) | 61.0 (−2.9) | 99.7 (−0.3) |
| + DeepAugment+AugMix [14] | 53.6 | 53.2 | 96.1 |
| + TTA | 55.2 (+1.6) | 51.0 (−2.2) | 93.5 (−2.6) |
| + Single point BN | 51.3 (−2.3) | 51.2 (−2.0) | 95.4 (−0.7) |
| + MEMO (ours) | **49.8** (−3.8) | **49.2** (−4.0) | 94.8 (−1.3) |
| + BN ($N = 256, n = 256$) | 45.4 (−8.2) | 48.8 (−4.4) | 96.8 (+0.7) |
| + Tent (online) | **43.5** (−10.1) | **46.9** (−6.3) | 96.7 (+0.6) |
| + Tent (episodic) | 47.1 (−6.5) | 50.1 (−3.1) | 96.6 (+0.5) |
| + MoEx+CutMix [24] | 74.8 | 64.5 | 91.9 |
| + TTA | 75.7 (+0.9) | 62.7 (−1.8) | 89.5 (−2.4) |
| + Single point BN | 71.0 (−3.8) | 62.6 (−1.9) | 91.1 (−0.8) |
| + MEMO (ours) | 69.1 (−5.7) | 59.4 (−3.3) | **89.0** (−2.9) |
| + BN ($N = 256, n = 256$) | 60.9 (−13.9) | 61.6 (−2.9) | 93.9 (+2.0) |
| + Tent (online) | 54.0 (−20.8) | 58.7 (−5.8) | 94.4 (+2.5) |
| + Tent (episodic) | 66.2 (−8.6) | 63.9 (−0.6) | 94.7 (+2.8) |
| RVT*-small [30] | 49.4 | 52.3 | 73.9 |
| + TTA | 53.0 (+3.6) | 49.0 (−3.3) | **68.9** (−5.0) |
| + Single point BN | 48.0 (−1.4) | 51.1 (−1.2) | 74.4 (+0.5) |
| + MEMO (ours) | **40.6** (−8.8) | **43.8** (−8.5) | 69.8 (−4.1) |
| + BN ($N = 256, n = 256$) | 44.3 (−5.1) | 51.0 (−1.3) | 78.3 (+4.4) |
| + Tent (online) | 46.8 (−2.6) | 50.7 (−1.6) | 82.1 (+8.2) |
| + Tent (adapt all) | 44.7 (−4.7) | 74.1 (+21.8) | 81.1 (+7.2) |

We summarize results for ImageNet-C, ImageNet-R, and ImageNet-A in Table 2, with complete ImageNet-C results in Appendix C. We again use indentations to indicate composition, e.g., the best results on ImageNet-C for our setting are attained through a combination of starting from a model trained with DeepAugment and AugMix [14] and using MEMO on top. For both ImageNet-C and ImageNet-R, and for both the ResNet-50 and RVT*-small models, combining MEMO with robust training techniques leads to new state-of-the-art performance among methods that observe only one test point at a time. We highlight in gray the methods that require multiple test points for adaptation, and we list in bold the best results from these methods which outperform the test time robustness methods. As Table 2 and prior work both show [40, 46], accessing multiple test points can be powerful for benchmarks such as ImageNet-C and ImageNet-R, in which inferred statistics

from the test input distribution may aid in prediction. However, these methods do not help, and oftentimes even hurt, for ImageNet-A. Furthermore, we find that these methods are less effective with the RVT*-small model, which may indicate their sensitivity to model architecture choices. Therefore, for this model, we also test a modification of Tent which adapts all parameters, and we find that this version of Tent works better for ImageNet-C but is significantly worse for ImageNet-R.

MEMO also results in substantial improvement for ImageNet-A. No prior test time adaptation methods have reported improvements on ImageNet-A, and some have reported explicit negative results [40]. As discussed, it is reasonable for adaptation methods that rely on multiple test points to achieve greater success on other benchmarks such as ImageNet-C, in which a batch of inputs provides significant information about the specific corruption that must be dealt with. In contrast, ImageNet-A does not have such obvious characteristics associated with the input distribution, as it is simply a collection of images that are difficult to classify. As MEMO instead extracts a learning signal from single test points, it is, to the best of our knowledge, the first test time adaptation method to report successful results on this testbed. We view the consistency with which MEMO outperforms the best prior methods, which change across different test sets, as a major advantage of the proposed method.

TTA is the most competitive prior method for ImageNet-A, e.g., it results in larger improvements than MEMO for the RVT*-small model. MEMO, however, achieves state-of-the-art performance among ResNet-50 models. To further compare MEMO to TTA, in Table 3, we evaluate whether MEMO can successfully adapt ResNext-101 models [50] and further improve performance on this challenging test set. We evaluate both a ResNext-101 (32x8d) baseline model pretrained on ImageNet, as well as the same model pretrained with weakly supervised learning (WSL) on billions of Instagram images [28]. For the

Table 3: ImageNet-A results for the ResNext-101s.

|  | ImageNet-A Error (%) |
|---|---|
| ResNext-101 [50] | 90.0 |
| + TTA | 83.2 (−6.8) |
| + Single point BN | 88.8 (−1.2) |
| + MEMO (ours) | 84.3 (−5.7) |
| + WSL [28] | 54.9 |
| + TTA | 49.1 (−5.8) |
| + Single point BN | 58.9 (+4.0) |
| + MEMO (ours) | **43.2** (−**11.7**) |

WSL model, we did not use single point BN adaptation for MEMO as we found this technique to be actually harmful to performance, and this corroborates previous findings [40]. From the results, we can see that, although both TTA and MEMO significantly improve upon the baseline model evaluation, MEMO ultimately achieves the best accuracy by a significant margin as it is more successful at adapting the WSL model. This suggests that MEMO may synergize well with large scale pretraining, and further exploring this combination is an interesting direction for future work.

### 4.2 Ablative Study

MEMO uses both adaptation and augmentations. In this section, we ablate the adaptation procedure and the number of augmentations, and in Appendix B we ablate the choice of augmentations.

**Adaptation procedure.** From the results above, we conclude that adaptation generally provides additional benefits beyond simply using TTA to predict via the marginal output distribution $\bar{p}_\theta(y|\mathbf{x})$. However, we can disentangle two distinct self-supervised learning signals that may be effective for adaptation: encouraging invariant predictions across different augmentations of the test point, and encouraging confidence via entropy minimization. The marginal entropy objective in Eq. 2 encapsulates both of these learning signals, but it cannot easily be decomposed into these pieces. We instead use two ablative adaptation methods that each only make use of one of these learning signals.

First, we consider optimizing the pairwise cross entropy between each pair of augmented points, i.e.,

$$\ell_{\text{PCE}}(\theta; \mathbf{x}) \triangleq \frac{1}{B \times (B-1)} \sum_{i=1}^{B} \sum_{j \neq i} H(p_\theta(\cdot|\tilde{\mathbf{x}}_i), p_\theta(\cdot|\tilde{\mathbf{x}}_j)),$$

Where $\tilde{\mathbf{x}}_i$ again refers to the $i$-th sampled augmentation applied to $\mathbf{x}$. Intuitively, this loss function encourages the model to adapt such that it produces the same predictive distribution for all augmentations of the test point, but it does not encourage the model to produce confident predictions.

Table 4: Ablating the adaptation objective to test pairwise cross entropy and conditional entropy (CE) based adaptation. MEMO generally performs the best, indicating that both encouraging invariance across augmentations and confidence are helpful in adapting the model.

|  | CIFAR-10 Error (%) | CIFAR-10.1 Error (%) | CIFAR-10-C Average Error (%) |
|---|---|---|---|
| ResNet-26 [11] | 9.2 | 18.4 | 22.5 |
| + MEMO (ours) | **7.3** $(-1.9)$ | **14.7** $(-3.7)$ | **19.6** $(-2.9)$ |
| $-\ell$ (Eq. 2) $+ \ell_{\text{PCE}}$ | 7.6 $(-1.6)$ | 15.3 $(-3.1)$ | 20.0 $(-2.5)$ |
| $-\ell$ (Eq. 2) $+ \ell_{\text{CE}}$ | 7.6 $(-1.6)$ | **14.7** $(-3.7)$ | 20.0 $(-2.5)$ |
|  | ImageNet-C mCE $\downarrow$ | ImageNet-R Error (%) | ImageNet-A Error (%) |
| RVT$^*$-small [30] | 49.4 | 52.3 | 73.9 |
| + MEMO (ours) | **40.6** $(-8.8)$ | **43.8** $(-8.5)$ | 69.8 $(-4.1)$ |
| $-\ell$ (Eq. 2) $+ \ell_{\text{CE}}$ | 41.2 $(-8.2)$ | 44.2 $(-8.1)$ | **69.7** $(-4.2)$ |

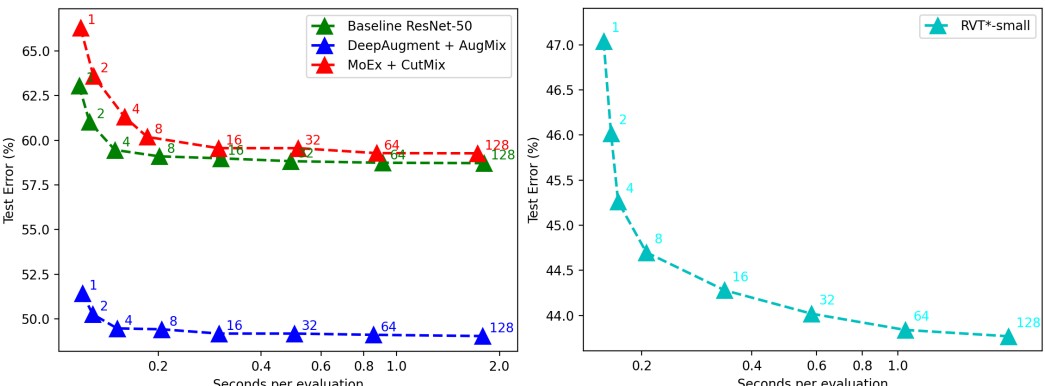

Figure 3: Plotting MEMO efficiency as seconds per evaluation (x axis) and % test error on ImageNet-R (y axis) for the ResNet-50 models (left) and RVT$^*$-small (right) while varying $B = \{1, 2, 4, 8, 16, 32, 64, 128\}$. Note the log scale on the x axis.

Conversely, as an objective that encourages confidence but not invariance, we also consider optimizing the conditional entropy objective detailed in Eq. 3. This ablation is effectively a version of the episodic variant of Tent [46] that produces augmented copies of a single test point rather than assuming access to a test batch. We first evaluate these ablations on the CIFAR-10 test sets. We use the same adaptation procedure and hyperparameters, with $\ell$ replaced with the above objectives.

The results are presented in Table 4. We see that MEMO, i.e., marginal entropy minimization, generally performs better than adaptation with either of the alternative objectives. This supports the hypothesis that both invariance across, and confidence on, the augmentations are important learning signals for self-supervised adaptation. When faced with CIFAR-10.1, we see poor performance from the pairwise cross entropy based adaptation method. On the original CIFAR-10 test set and CIFAR-10-C, the ablations perform nearly identically and uniformly worse than MEMO. To further test the $\ell_{\text{CE}}$ ablation, which is the stronger of the two ablations, we also evaluate it on the ImageNet test sets for the RVT$^*$-small model. We find that, similarly, minimizing conditional entropy generally improves performance compared to the baseline evaluation. MEMO is more performant for ImageNet-C and ImageNet-R. Adaptation via $\ell_{\text{CE}}$ performs slightly better for ImageNet-A, though for this problem and model, TTA is still the best method. Thus, MEMO results in relatively small, but consistent, performance gains compared to only maximizing confidence on the augmentations.

**Number of augmentations.** In Figure 3, we analyze the % test error of MEMO adaptation on ImageNet-R as a function of the efficiency of adaptation, measured in seconds per evaluation. We achieve various tradeoffs by varying the number of augmented copies $B =$

$\{1, 2, 4, 8, 16, 32, 64, 128\}$. We note that small values of $B$ such as $4$ and $8$ can already provide significant performance gains, thus a practical tradeoff between efficiency and accuracy is possible.

For large $B$, the wall clock time is dominated by computing the augmentations. For the baseline ResNet-50 model, single point BN adaptation requires an average of $0.0252$ seconds per test point. TTA and MEMO with $B = 64$ are much slower – $0.7742$ and $0.9746$ seconds, respectively, per test point – but can be made significantly more efficient by using $B = 4$ augmented samples – $0.0631$ and $0.1037$ seconds, respectively. In our implementation, we do not compute augmentations in parallel, though in principle this is possible for AugMix and should drastically improve efficiency overall. These experiments used four Intel Xeon Skylake 6130 CPUs and one NVIDIA TITAN RTX GPU.

## 5   Discussion

We presented MEMO, a method for test time robustification again distribution shift via adaptation and augmentation. MEMO does not require access or changes to the model training procedure and is thus broadly applicable for a wide range of model architectures pretrained in a number of different ways. Furthermore, MEMO adapts at test time using single test inputs, thus it does not assume access to multiple test points as in several recent methods for test time adaptation [40, 46]. On a range of CIFAR-10 and ImageNet distribution shift benchmarks, and for ResNet, vision transformer, and, to an extent, ResNext models, MEMO consistently improves performance at test time and achieves several new state-of-the-art results for these models in the single test point setting.

Inference via MEMO is more computationally expensive than standard model inference due to its augmentation and adaptation procedure – though, as the experiments above show, more favorable tradeoffs between efficiency and accuracy are possible with smaller values of $B$, the number of augmentations per test point. One interesting direction for future work is to develop techniques for selectively determining when to adapt the model in order to achieve more efficient inference. For example, with well calibrated models [10], we may run simple "feedforward" inference when the prediction confidence is over a certain threshold, thus achieving better efficiency. Additionally, it would be interesting to explore MEMO in the test setting where the model is allowed to continually adapt as more test data is observed. In our preliminary experiments in this setting, MEMO tended to lead to degenerate solutions, e.g., the model predicting a constant label with maximal confidence regardless of the input. This failure mode may potentially be rectified by carefully choosing which parameters to adapt, such as only adapting the parameters in BN layers [46], or regularizing the model such that it does not change too drastically from the pretrained model [26].

## Acknowledgments and Disclosure of Funding

We thank members of the Robotic AI and Learning Lab and Berkeley AI Research for helpful discussions and feedback. MZ was supported in part by an NDSEG fellowship. CF is a CIFAR fellow. This research was partially supported by ARL DCIST CRA W911NF-17-2-0181 and ARO W911NF-21-1-0097.

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
