# A  Experimental Protocol

We selected hyperparameters using the four disjoint validation corruptions provided with CIFAR-10-C and ImageNet-C [12]. As the other benchmarks are only test sets and do not provide validation sets, we used the same hyperparameters found using the corruption validation sets and do not perform any additional tuning. We considered the following hyperparameters when performing a grid search.

- Learning rate $\eta$: $10^{-3}$, $10^{-4}$, $10^{-5}$, $10^{-6}$; then, $5\times$, $2.5\times$, and $0.5\times$ the best value.
- Number of gradient steps: 1, 2.
- % of the maximum loss value to threshold for adaptation: 50, 100.
- Prior strength $N$: 8, 16, 32.

Beyond learning rate and number of gradient steps, we also evaluated using a simple "threshold" by performing adaptation only when the marginal entropy was greater than $50\%$ of the maximum value ($\log 1000$ for ImageNet-C), though we found that this resulted in slightly worse validation performance. We also considered different values of the prior strength $N$ for single point BN adaptation, and we found that 16 performed best on the validation sets as suggested in Schneider et al. [40]. For the ResNet models, we use stochastic gradients as the update rule $G$; for ResNet-26 models, we set the number of augmentations $B = 32$ and the learning rate $\eta = 0.005$; and for ResNet-50 models, we set $B = 64$ and $\eta = 0.00025$.

For the RVT$^*$-small, we additionally considered the following hyperparameters in our grid search.

- Update rule $G$: stochastic gradients, Adam [20, 27].
- Weight decay: 0.0, 0.1.
- Prior strength $N$: 1, 2, 4, 8, 16, 32, $\infty$ (i.e., no single point BN adaptation).

Based on the ImageNet-C validation sets, we use AdamW [27] as the update rule $G$, with learning rate $\eta = 0.00001$ and weight decay 0.01, and $B = 64$. We use the same hyperparameters for the ResNext-101 models without any additional tuning, except we use $B = 32$ due to memory limits.

In the CIFAR evaluation, we compare to TTT, which, as noted, can also be applied to single test inputs but requires a specialized training procedure [44]. Thus, the ResNet-26 model we use for our method closely follows the modifications that Sun et al. [44] propose, in order to provide a fair point of comparison. In particular, Sun et al. [44] elect to use group normalization [49] rather than BN, thus single point BN adaptation is not applicable for this model architecture. As noted before, TTT also requires the joint training of a separate rotation prediction head, thus further changing the model architecture, while MEMO directly adapts the standard pretrained model.

The TTA results are obtained using the same AugMix augmentations as for MEMO. The single point BN adaptation results use $N = 16$, as suggested by Schneider et al. [40]. As noted, the BN adaptation results (using multiple test points) are obtained using $N = 256$ as the prior strength and batches of 256 test inputs for adaptation. For Tent, we use the hyperparameters suggested in Wang et al. [46]: stochastic gradients with learning rate 0.00025 and momentum 0.9. The adaptation is performed with test batches of 64 inputs – for the online version of Tent, prediction and adaptation occur simultaneously and the model is allowed to continuously adapt through the entire test epoch. Since Wang et al. [46] did not experiment with transformer models, we also attempted to run Tent with Adam [20] and AdamW [27] and the various hyperparameters detailed above for the RVT$^*$-small model; however, we found that this generally resulted in worse performance than using stochastic gradient updates.

We obtain the baseline ResNet-50 and ResNext-101 (32x8d) parameters directly from the `torchvision` library.
The parameters for the ResNet-50 trained with DeepAugment and AugMix are obtained from
`https://drive.google.com/file/d/1QKmc_p6-qDkh51WvsaS9HKFv8bX5jLnP`.
The parameters for the ResNet-50 trained with moment exchange and CutMix are obtained from
`https://drive.google.com/file/d/1cCvhQKV93pY-jj8f5jITywkB9EabiQDA`.
The parameters for the small robust vision transformer (RVT$^*$-small) model are obtained from
`https://drive.google.com/file/d/1g4OhuqDVthjS2H5sQV3ppcfcWEzn9ekv`.
The parameters for the ResNext-101 (32x8d) trained with WSL are obtained from
`https://download.pytorch.org/models/ig_resnext101_32x8-c38310e5.pth`.

Table 5: Evaluating the episodic version of Tent with a batch size of 1, which corresponds to a simple entropy minimization approach for the test time robustness setting. This approach also uses single point BN adaptation, and entropy minimization does not provide much additional gain.

| | ImageNet-C mCE ↓ | ImageNet-R Error (%) | ImageNet-A Error (%) |
|---|---|---|---|
| Baseline ResNet-50 [11] | 76.7 | 63.9 | 100.0 |
| + Single point BN | 71.4 (−5.3) | 61.1 (−2.8) | 99.4 (−0.6) |
| + MEMO (ours) | 69.9 (−6.8) | 58.8 (−5.1) | 99.1 (−0.9) |
| + Tent (episodic, batch size 1) [46] | 70.4 (−6.3) | 60.0 (−3.9) | 99.3 (−0.7) |
| + DeepAugment+AugMix [14] | 53.6 | 53.2 | 96.1 |
| + Single point BN | 51.3 (−2.3) | 51.2 (−2.0) | 95.4 (−0.7) |
| + MEMO (ours) | **49.8** (−**3.8**) | **49.2** (−**4.0**) | 94.8 (−1.3) |
| + Tent (episodic, batch size 1) | 50.7 (−2.9) | 50.7 (−2.5) | 95.2 (−0.9) |
| + MoEx+CutMix [24] | 74.8 | 64.5 | 91.9 |
| + Single point BN | 71.0 (−3.8) | 62.6 (−1.9) | 91.1 (−0.8) |
| + MEMO (ours) | 69.1 (−5.7) | 59.4 (−3.3) | **89.0** (−**2.9**) |
| + Tent (episodic, batch size 1) | 69.9 (−4.9) | 61.7 (−2.8) | 90.6 (−1.3) |
| RVT*-small [30] | 49.4 | 52.3 | 73.9 |
| + Single point BN | 48.0 (−1.4) | 51.1 (−1.2) | 74.4 (+0.5) |
| + MEMO (ours) | **40.6** (−**8.8**) | **43.8** (−**8.5**) | **69.8** (−**4.1**) |
| + Tent (episodic, batch size 1) | 47.9 (−1.5) | 50.9 (−1.4) | 74.4 (+0.5) |

Table 6: Ablating the augmentation functions to test standard augmentations (random resized cropping and horizontal flips). When changing the augmentations used, the post-adaptation performance generally does not change much, though it suffers the most on CIFAR-10-C.

| | CIFAR-10 Error (%) | CIFAR-10.1 Error (%) | CIFAR-10-C Average Error (%) |
|---|---|---|---|
| ResNet-26 [11] | 9.2 | 18.4 | 22.5 |
| + MEMO (ours) | 7.3 (−1.9) | 14.7 (−3.7) | **19.6** (−**2.9**) |
| − AugMix [13] + standard augs | **7.2** (−**2.0**) | **14.6** (−**3.8**) | 20.2 (−2.3) |

# B  Analysis on Augmentations

One may wonder: are augmentations needed in the first place? In the test time robustness setting when only one test point is available, how would simple entropy minimization fare? We answer this question in Table 5 by evaluating the episodic variant of Tent (i.e., with model resetting after each batch) with a test batch size of 1. This approach is also analogous to a variant of MEMO that does not use augmentations, since for one test point and no augmented copies, conditional and marginal entropy are the same. Similar to MEMO, we also incorporate single point BN adaptation with $N = 16$, in place of the standard BN adaptation that Tent typically employs using batches of test inputs. The results in Table 5 indicate that entropy minimization on a single test point generally provides marginal performance gains beyond just single point BN adaptation. This empirically shows that using augmentations is important for achieving the reported results.

We also wish to understand the importance of the choice of augmentation functions $\mathcal{A}$. As mentioned, we used AugMix [13] in the previous experiments as it best fit our criteria: AugMix requires only the input $\mathbf{x}$, and randomly sampled augmentations lead to diverse augmented data points. A simple alternative is to instead use the "standard" set of augmentations commonly used in ImageNet training, i.e., random resized cropping and random horizontal flipping. We evaluate this ablation of using MEMO with standard augmentations also on the CIFAR-10 test sets, again with the same hyperparameter values. From the results in Table 6, we can see that MEMO is still effective with simpler augmentation functions. This is true particularly for the cases where there is no test shift, as in the original CIFAR-10 test set, or subtle shifts as in CIFAR-10.1; however, for the more severe and systematic CIFAR-10-C shifts, using heavier AugMix data augmentations leads to greater

performance gains over the standard augmentations. Furthermore, this ablation was conducted using the ResNet-26 model, which was trained with standard augmentations – for robust models such as those in Table 2, AugMix may offer greater advantages at test time since these models were exposed to heavy augmentations during training.

## C  Full CIFAR-10-C and ImageNet-C Results

In the following tables, we present test results broken down by corruption and level for CIFAR-10-C for the methods evaluated in Table 1. We omit joint training and TTT because these results are available from Sun et al. [44]. Our test results for ImageNet-C are provided in the CSV files in the supplementary material.

Table 7: Test error (%) on CIFAR-10-C level 5 corruptions.

|  | gauss | shot | impul | defoc | glass | motn | zoom | snow | frost | fog | brit | contr | elast | pixel | jpeg |
|---|---|---|---|---|---|---|---|---|---|---|---|---|---|---|---|
| ResNet-26 | 48.4 | 44.8 | 50.3 | 24.1 | 47.7 | 24.5 | 24.1 | 24.1 | 33.1 | 28.0 | 14.1 | 29.7 | 25.6 | 43.7 | 28.3 |
| + TTA | 43.4 | 39.6 | 42.9 | 28.3 | 44.7 | 26.3 | 26.3 | 21.4 | 28.5 | 23.3 | 12.1 | 32.9 | 21.7 | 43.2 | 21.7 |
| + MEMO (ours) | 43.5 | 39.8 | 43.3 | 26.4 | 44.4 | 25.1 | 25.0 | 20.9 | 28.3 | 22.8 | 11.9 | 28.3 | 21.1 | 42.8 | 21.7 |

Table 8: Test error (%) on CIFAR-10-C level 4 corruptions.

|  | gauss | shot | impul | defoc | glass | motn | zoom | snow | frost | fog | brit | contr | elast | pixel | jpeg |
|---|---|---|---|---|---|---|---|---|---|---|---|---|---|---|---|
| ResNet-26 | 43.8 | 37.2 | 39.3 | 14.8 | 48.0 | 19.9 | 18.7 | 22.0 | 24.9 | 15.1 | 11.4 | 16.8 | 19.1 | 27.9 | 24.9 |
| + TTA | 39.5 | 32.0 | 31.8 | 15.4 | 45.0 | 20.9 | 20.2 | 18.9 | 21.7 | 12.9 | 9.3 | 16.8 | 17.7 | 25.7 | 18.9 |
| + MEMO (ours) | 39.7 | 32.3 | 32.2 | 14.7 | 45.0 | 20.0 | 19.2 | 18.7 | 21.1 | 12.5 | 9.3 | 15.2 | 16.9 | 25.2 | 18.9 |

Table 9: Test error (%) on CIFAR-10-C level 3 corruptions.

|  | gauss | shot | impul | defoc | glass | motn | zoom | snow | frost | fog | brit | contr | elast | pixel | jpeg |
|---|---|---|---|---|---|---|---|---|---|---|---|---|---|---|---|
| ResNet-26 | 40.0 | 33.8 | 26.4 | 11.5 | 37.3 | 20.0 | 16.6 | 20.0 | 24.7 | 12.2 | 10.5 | 13.6 | 15.0 | 18.4 | 22.7 |
| + TTA | 34.3 | 27.7 | 20.3 | 11.3 | 32.9 | 20.7 | 16.7 | 16.3 | 21.1 | 9.8 | 8.5 | 12.5 | 13.7 | 14.5 | 17.2 |
| + MEMO (ours) | 34.4 | 27.9 | 20.5 | 10.8 | 32.8 | 19.8 | 16.1 | 16.1 | 20.9 | 9.6 | 8.6 | 11.7 | 13.2 | 14.5 | 17.2 |

Table 10: Test error (%) on CIFAR-10-C level 2 corruptions.

|  | gauss | shot | impul | defoc | glass | motn | zoom | snow | frost | fog | brit | contr | elast | pixel | jpeg |
|---|---|---|---|---|---|---|---|---|---|---|---|---|---|---|---|
| ResNet-26 | 30.1 | 21.8 | 21.1 | 9.7 | 38.3 | 15.3 | 13.8 | 21.2 | 17.6 | 10.5 | 9.7 | 11.6 | 12.9 | 15.4 | 21.3 |
| + TTA | 25.3 | 16.9 | 15.8 | 8.5 | 33.8 | 15.3 | 13.7 | 17.8 | 14.5 | 8.7 | 7.8 | 10.0 | 11.0 | 12.0 | 16.1 |
| + MEMO (ours) | 25.3 | 16.9 | 15.9 | 8.4 | 33.5 | 14.6 | 13.0 | 17.7 | 14.4 | 8.5 | 7.7 | 9.6 | 10.7 | 11.9 | 16.2 |

Table 11: Test error (%) on CIFAR-10-C level 1 corruptions.

|  | gauss | shot | impul | defoc | glass | motn | zoom | snow | frost | fog | brit | contr | elast | pixel | jpeg |
|---|---|---|---|---|---|---|---|---|---|---|---|---|---|---|---|
| ResNet-26 | 20.8 | 16.5 | 15.8 | 9.2 | 38.9 | 11.8 | 12.8 | 13.9 | 13.4 | 9.7 | 9.4 | 9.6 | 13.1 | 12.0 | 16.4 |
| + TTA | 15.8 | 12.8 | 11.8 | 7.3 | 35.1 | 10.8 | 12.5 | 11.0 | 10.8 | 7.4 | 7.4 | 7.7 | 11.4 | 9.2 | 12.5 |
| + MEMO (ours) | 16.1 | 12.9 | 11.9 | 7.4 | 34.7 | 10.4 | 12.1 | 11.0 | 10.7 | 7.4 | 7.3 | 7.5 | 10.9 | 9.2 | 12.5 |