# OpenReview forum: "MEMO: Test Time Robustness via Adaptation and Augmentation"
_NeurIPS.cc/2022/Conference — NeurIPS 2022 Accept_

### Official Review · Reviewer_bMhD · 2022-07-10

**Rating:** 6
**Confidence:** 4
**Soundness:** 3 good
**Presentation:** 4 excellent
**Contribution:** 3 good

**Summary:**

This paper proposed an approach that performs test-time adaptation using one test point. Most past test-time adaptation work requires accessing multiple test points under the same distribution shifts. The method first augments the query test data with different augmentation instances from AugMix, and then use the classifier to estimate the predictions. Then the whole model backbone is adapted to produce invariant predictions across augmented versions of the test data, and also optimized to be confident on the prediction, which is different from past work where only BN layer is optimized. Experiment on ImageNet-A, C, R are presented. Ablations studies on the effect of invariance and confidence are presented.

**Questions:**

None

**Strengths And Weaknesses:**

Strengths:
1. This work performs test-time adaptation on single test point, which is more realistic than past work setup. It is a simple and effective approach.
2. The method is evaluated over more than 4 OOD benchmark, and demonstrates improvement on ImageNet-A for the first time.
3. The ablation study shows the effect of each component of the proposed approach.
4. Presentation is clear.

Weakness:
1. What is the inference time the test time adaptation method spends compared with non-adaptation method and other test-time adaptation method? A Table should be provided.
2. Missing related work: [1] uses test time adaptation on augmented input images to defend adversarial distribution shifts.


[1] Mao et al. Adversarial Attacks are Reversible with Natural Supervision. ICCV 2021.  https://arxiv.org/abs/2103.14222

---

> ### Author Response · Authors · 2022-07-30
> **Response to Reviewer bMhD**
>
> Thank you for your helpful suggestions. Here, we summarize a few changes we have made to the paper in line with your comments. Should you have any outstanding questions or comments, please let us know and we are happy to respond.
>
> > “What is the inference time the test time adaptation method spends compared with non-adaptation method and other test-time adaptation method?”
>
> We have added some approximate inference times for TTA, single point BN, and MEMO to Appendix B.2 of the supplementary material. For the baseline ResNet-50 model and ImageNet-R, these methods require an average of 0.7742, 0.0252, and 0.9746 seconds, respectively, per test point. Note that our implementation does not parallelize the augmentations per sample, but doing so should significantly improve the efficiency of both TTA and MEMO. Indeed, if we reduce the number of augmented samples per test point from 64 to 4, TTA and MEMO then require an average of 0.0631 and 0.1037 seconds, respectively, per test point. Note that Figure 3 in Appendix B.2 shows that MEMO maintains most of the performance gains even with as few as 4 augmented samples per test point.
>
> > “Missing related work”
>
> Thank you for pointing out this paper, we have added it to Section 2.

---

> > ### Comment · Reviewer_bMhD · 2022-08-09
> > **My questions are adressed.**
> >
> > Thanks for the authors' response.

---

### Official Review · Reviewer_yhos · 2022-07-11

**Rating:** 5
**Confidence:** 4
**Soundness:** 3 good
**Presentation:** 3 good
**Contribution:** 2 fair

**Summary:**

The paper proposes a method for single-point test-time adaptation, i.e., test-time adaptation without an access of multiple i.i.d. samples in test-time, by (a) using a set of data augmentation techniques instead to build a batch to adapt, and by (b) minimizing the marginal entropy of the predictions from the batch of (augmented) samples. Experimental results on CIFAR and ImageNet, along with a diverse set of distribution shift benchmarks - CIFAR-10-C, CIFAR-10.1 and ImageNet-C/R/A and architectures covering both convolutional networks and vision transformer, show that the proposed method can outperform previous single-point adaptation methods.

**Questions:**

- The paper states that MEMO adapts every parameters in the model unlike previous BN-adaptation based methods. But how does MEMO perform under BN-only adaptation? This would give a clearer comparison with Single-point BN baseline.
- Given that MEMO uses augmentations from AugMix, I wonder how the paper handled in their evaluation for corruptions that are in common to the AugMix augmentations - some corruption types in CIFAR-10-C may share some semantics with some types of augmentations in AugMix, which means that the method thus may leveraged an additional prior knowledge of augmentations compared to baselines, i.e., TTA and Single-point BN.

**Limitations:**

The paper does include an adequate discussion for the limitations, but apparently does not for the potential societal impacts.

**Strengths And Weaknesses:**

**Strengths**

- The paper is clearly-written and easy-to-follow.
- The proposed method is simple and easy-to-use.
- The paper performs extensive experiments across diverse datasets and architectures to show the effectiveness.
- The paper tackles a challenge that has been crucial in the literature of test-time adaptation, i.e., the requirement of batch information.

**Weaknesses**

- The overall gains can be marginal considering its additional computational costs: although the paper demonstrates that the proposed MEMO could outperform two main baselines, i.e., TTA and Single-point BN, the performance gap between MEMO and Single-point BN is overall quite small while MEMO has several times more inference time in my understanding.
- The ablation study is inconclusive on identifying which part of the method is the most crucial to improve results. I think currently several important ablations are missing in the ablation study: e.g., MEMO without Single-BN adaptation, more extensive ablations on the data augmentations used, # of augmented samples used per inference.
- Regarding the reported ablations (in Table 4), on the other hand, I found the performance of the ablation "$+ \ell_{\mathrm{CE}}$" works quite similarly with the proposed marginal entropy based loss, which means that a Tent variant is actually performs similarly to MEMO, and the actual form adaptation objective does not much matter. This can decrease the technical novelty of the paper.

---

> ### Author Response · Authors · 2022-07-30
> **Initial Response to Reviewer yhos**
>
> Thank you for your helpful suggestions. We have updated the paper in line with your comments and questions, and here we summarize several important clarifications we have made. Please let us know if these changes help to address your concerns, we look forward to your response.
>
> > “overall gains can be marginal considering its additional computational costs… the performance gap between MEMO and Single-point BN is overall quite small while MEMO has several times more inference time”
>
> We agree that, in some cases, MEMO only narrowly outperforms single point BN. However, for ImageNet-A and the robust vision transformer model, the performance gap is larger. In these settings, single point BN leads to very little improvement over just standard inference (Table 2). It is reasonable to expect that no method would be useful in every possible setting. However, as you have noted, we have conducted “extensive experiments across diverse datasets and architectures” and MEMO demonstrates consistent performance gains over the next best methods. Despite the fact that individual gains are sometimes small, which we have acknowledged in the paper at the end of Section 4, we view this consistency as a major advantage of MEMO. This consistency does indeed come at the cost of increased inference time, though this can be mitigated to some degree as discussed next.
>
> > missing ablation: “# of augmented samples used per inference”
>
> We already have this ablation in Appendix B.2 in the original supplementary material. To briefly summarize, performance improves monotonically with the number of augmented samples, but most of the gains are realized using just 4 to 8 augmented samples, which represents a favorable tradeoff between performance and efficiency of inference. We have better highlighted this result in the first part of Section 4.
>
> > “I wonder how the paper handled in their evaluation for corruptions that are in common to the AugMix augmentations”
>
> The AugMix authors state, on page 4 of their paper [1], that they “exclude [augmentations] which overlap with ImageNet-C corruptions”, thus there are no corruptions that we used in our evaluation that are also part of AugMix. We have clarified this point in Section 3.2.
>
> [1] https://arxiv.org/abs/1912.02781
>
> > “I think currently several important ablations are missing”
>
> In preliminary experiments, we found that both “MEMO without single-BN adaptation” and MEMO with “BN-only adaptation” performed worse than standard MEMO, though they still generally improved performance. We will aim to add these ablations, but these results will not be ready until later on in the discussion phase. We will share these results as soon as they are ready, and we expect that they will align with the preliminary results we obtained.
>
> Regarding “more extensive ablations on the data augmentations used”, we used AugMix as our default because it satisfies our desiderata (diverse augmentations applied directly to the input image). Though we do not believe that changing the augmentations will significantly affect the results, we are happy to include a more extensive ablation study of data augmentations if you can provide additional details on what you wish to see.

---

> ### Author Response · Authors · 2022-08-08
> **Followup Response to Reviewer yhos -- Additional Ablations, Response Requested**
>
> We are following up with some results from the requested ablations: “MEMO without single-BN adaptation” and MEMO with “BN-only adaptation”. As the discussion phase is nearly over, we are kindly requesting a timely response.
>
> | | ImageNet-R Error (%) | ImageNet-A Error (%) |
> | :--- | ---: | ---: |
> | Baseline ResNet-50 | 63.9 | 100.0 |
> | MEMO without single point BN | 61.2 | 99.5 |
> | MEMO with BN only adaptation | 60.4 | 99.3 |
> | MEMO (ours) | 58.8 | 99.1 |
> | | | |
> | DeepAugment + AugMix | 53.2 | 96.1 |
> | MEMO without single point BN | 51.2 | 95.6 |
> | MEMO with BN only adaptation | 50.7 | 95.2 |
> | MEMO (ours) | **49.2** | 94.8 |
> | | | |
> | MoEx + CutMix | 64.5 | 91.9 |
> | MEMO without single point BN | 61.9 | 91.3 |
> | MEMO with BN only adaptation | 61.2 | 90.6 |
> | MEMO (ours) | 59.4 | **89.0** |
> | | | |
> | | | |
> | RVT*-small | 52.3 | 73.9 |
> | MEMO without single point BN | 45.0 | 70.0 |
> | MEMO with BN only adaptation | 50.9 | 74.2 |
> | MEMO (ours) | **43.8** | **69.8** |
>
> We do not yet have results for ImageNet-C, as this test set is several orders of magnitude larger than ImageNet-R and ImageNet-A. However, these results that we have obtained thus far agree with the results from our preliminary experiments. Specifically, these alternate versions of MEMO are generally still able to improve performance compared to the standard test time inference procedure, but they are not as effective as the proposed version of MEMO.
>
> Should you have other questions or concerns, we would be happy to try and provide another response in the time remaining. Otherwise, we would like to know if your concerns have been adequately addressed, and we thank you again for your helpful feedback.

---

> > ### Comment · Reviewer_yhos · 2022-08-08
> > **Response to the rebuttal**
> >
> > Thanks for addressing my concerns and questions. Especially, I appreciate the additional experiments performed to improve ablation study, and believe that incorporating these results in the final draft would strengthen the paper. I have raised my score given that many of the concerns of mine are addressed now, although I still think the marginal gains compared to "Single-point BN" (except for RVTs) can be a weakness.

---

### Official Review · Reviewer_dZZX · 2022-07-11

**Rating:** 7
**Confidence:** 4
**Soundness:** 3 good
**Presentation:** 4 excellent
**Contribution:** 2 fair

**Summary:**

## Post rebuttal
I'm bumping up my rating to Accept.

## Original
This paper proposes a method for test-time adaptation for a single sample by augmenting the data point and reducing the marginal entropy over these augmented inputs. The experiments shown on Imagenet-C, -A, -R and CIFAR-10, 10-C, 10.1 show that MEMO is capable to reducing test errors with access to a single point.

**Questions:**

* How does Memo fare with batches? I see single point BN results, and Tent with batch size 1 (in Table 5) but it might be interesting to see how Memo does with access to mini-batches comparable to the ones used in Tent.
* Why do the authors use AugMix specifically? Are there any specific properties of Augmix that are useful for the proposed method, since they show in Table 6 that using standard augs gets very close ($\approx$ 0.5\%)?

**Limitations:**

Limitations of the method would be limited technical novelty, but has been more than made up through extensive experiments. Also a theoretical understanding of the method is completely lacking, though I understand this might warrant its own paper.

**Strengths And Weaknesses:**

### Strengths:
The paper is clearly written, and well motivated. The ablations are shed further light on the importance of each component of the method. The results on Imagenet-A are especially impressive.

### Weakness:

* The method is a black-box. Why does encouraging same prediction across augmentations  necessarily leads to better results?
* The paper is missing references to several new paper on test-time adaptation [1, 2] etc. Several improvements to Tent have been proposed, and comparison to those is needed to place the importance of the current paper in the latest literature. Additionally, a method [3] that is quite similar to the current one has not been discussed.
* A discussion on what architectural choices lead to better results is missing i.e., improvements over Tent are small (in most cases), and all the architectures use some normalization layer. How would the authors attribute their method's successes to each kind of network layer? Would the method be effective on networks without normalization?
* The paper can be expanded by experiments on various other shifts (domain adaptation problems), or tasks like semantic segmentation (as shown in Tent).


[1] Test-time classifier adjustment module for model-agnostic domain generalization
[2] MT3: Meta Test-Time Training for Self-Supervised Test-Time Adaption
[3] Test time Adaptation through Perturbation Robustness https://openreview.net/forum?id=GbBeI5z86uD

---

> ### Author Response · Authors · 2022-07-30
> **Response to Reviewer dZZX**
>
> Thank you for your helpful suggestions. Here, we summarize a few changes we have made to the paper in line with your comments, and we also address your questions to the best of our ability. Should you have any outstanding questions or comments, please let us know and we are happy to respond.
>
> > “The paper is missing references to several new paper on test-time adaptation”
>
> Thank you for pointing out [2, 3], we have added these to Section 2. Please note that we have already cited [1] in Section 1 of the original paper, and we now also reference this work in Section 2. The methods proposed in [1, 2] study the settings in which multiple test points are available for adaptation or the training procedure can be modified, respectively. Thus, we do not view these methods as core comparisons to our work as we are focused on the setting in which we adapt pretrained models using only one test point.
>
> > “improvements over Tent are small (in most cases)”
>
> Tent uses batches or even entire datasets of test points for adaptation, whereas MEMO uses only a single test point. Thus, we include the Tent comparison more to illustrate the fact that batch information is oftentimes very useful, rather than to claim that MEMO performs better than Tent (which, generally, we cannot claim). We view the core contribution of our work as proposing a method that works consistently across a wide range of datasets and models while making fewer assumptions about the training and test settings compared to prior methods.
>
> > “A discussion on what architectural choices lead to better results is missing”
>
> We generally find that, although not strictly necessary for MEMO, batch normalization (BN) layers are useful because we can then combine MEMO with single point BN adaptation. We have better highlighted this in Section 3.2. We have not closely studied other architectural choices, as we have instead focused on using state-of-the-art pretrained models, since this is likely of the most interest to the community. Our results generally indicate that MEMO is readily compatible with multiple different state-of-the-art model architectures. To the best of our knowledge, it is rare for modern deep networks to not have any form of normalization layer, thus we do not consider this in our work.
>
> > “​​Why does encouraging same prediction across augmentations necessarily leads to better results?”
>
> As you have noted, analyzing this theoretically is out of scope for this work and “might warrant its own paper”. But we believe that encouraging invariance is, in general, well motivated from the perspective of reducing the model’s reliance on irrelevant features. I.e., different augmented samples may have different spurious cues, but all (or most) augmented inputs will still have the same “causal” cue that indicates the correct label, and MEMO encourages the model to pay the most attention to this causal cue. We view this as an interesting direction for future study.
>
> > “Why do the authors use AugMix specifically?”
>
> As you have correctly pointed out, AugMix performed slightly better in our preliminary experiments compared to more standard augmentations, likely because it yields more diverse augmented samples. Thus, we chose AugMix over standard augmentations. We also tried a “patch-wise” augmentation for the robust vision transformer [4], though we found this to also be slightly worse than AugMix. One property that is useful for MEMO, that these augmentations all satisfy but some others don’t, is that they are applied directly to the input image.
>
> [4] Mao et al, “Towards Robust Vision Transformer”. CVPR ‘22.

---

> > ### Comment · Reviewer_dZZX · 2022-08-05
> > **Response to the rebuttal**
> >
> > Thank you for the clarifications.
> >
> > After reading the other reviews, and the author rebuttals, I'm bumping up my rating to Accept. There are unanswered questions about why the proposed method works, but as the authors remark, it might warrant its own investigations.

---

### Official Review · Reviewer_iZWy · 2022-07-18

**Rating:** 7
**Confidence:** 3
**Soundness:** 4 excellent
**Presentation:** 4 excellent
**Contribution:** 3 good

**Summary:**

- This work presents a simple, empirically-driven approach (MEMO) for test-time adaptation of arbitrary parametric deep models, given only a single test point. The approach is modular, and the authors examine it in conjunction with other relevant approaches in a wide test suite.
- The approach centers on augmenting the given test point before evaluating the model on all augmentations, then applying an entropy minimization loss on the average of all evaluations' prediction vectors, and taking a single gradient step to update the model. The procedure is well-described in Algorithm 1 and section 3.1.

The flow of the work is to:
- Introduced related work.
- Introduce the MEMO procedure and motivate the associated loss function.
- Describe auxiliary approaches that can be combined with MEMO.
- Examine error rates across different datasets when evaluating with MEMO, other adaptation approaches, and combinations of approaches.
- Ablate MEMO in several ways and show that both of the loss' targets (confidence and agreement) are necessary.

**Questions:**

Questions:
- I have no outstanding questions regarding this work. Any such questions (e.g. questions about the effects of variation in the augmentation pools) are answered in the appendices. I believe that the authors are generally aware of the effects and limitations of their work, and communicate its scope well and honestly.


**Limitations:**

Limitations:
- The authors argue that their method presents small, but consistent gains, which is true. They do not overstate the empirical result.
- In the ImageNet-A problem on RVT*-small model, the MEMO approach does not outperform other adaptation methods. The authors correctly call this out.
- The authors correctly call out and demonstrate that access to larger numbers of test points at once can improve adaptation in some settings, and argue that their method only represents SOTA gains on certain datasets _under the restriction of accessing a single test point for adaptation._

- This paper's scope is largely isolated from societal impacts, and I do not believe a discussion of them is needed.

**Strengths And Weaknesses:**

Strengths:
- The approach is, to my knowledge, novel, although it draws inspiration from a variety of works on data augmentation and test-time adaptation.
- The approach is well-communicated, with Figure 1 and Algorithm 1 capturing the key details. The choice of loss function is clearly motivated in 3.1 as optimizing for confidence of prediction as well as agreement between predictions on different augmentations.
- The approach presents impactful improvements for test-time adaptation on variants of ImageNet	with ResNet-50, as well as on ImageNet-A on ResNext-101.
- The ablative study split across the main paper and appendices is extensive, ablating not only each part of the loss objective (agreement and confidence), but also the choice of augmentations (Appendix B), as well as the number of augmentations sampled for evaluation (Appendix B).
- Communication is clear.

Weaknesses:
- Improvements on variants of CIFAR-10 are slim. This is not an issue with the paper or approach, so much as an observation that MEMO is not a panacea and it is more useful in some settings than others.
- This work is not, in my view, a new literature paradigm. It combines several established techniques (augmentation, test-time adaptation, entropy minimization) into a novel module to add onto other models and robustification approaches, in order to induce performance gains. Most of the interesting future directions this work opens (and addresses in Discussion section) are further improvements on the MEMO procedure, as opposed to broadly new ways to think about the problem of test-time adaptation.

---

> ### Author Response · Authors · 2022-07-30
> **Response to Reviewer iZWy**
>
> Thank you for your thorough and careful review. We generally agree with all of your points made. Should you have any outstanding questions, please let us know and we are happy to answer them as best as we can.

---

> > ### Comment · Reviewer_iZWy · 2022-08-05
> > **Response to Rebuttal**
> >
> > Thank you for responding to other reviewers' concerns.
> > I am satisfied with the rebuttal and will maintain my original score.

---

### Public Comment · ~Ebrahim_Feghhi1 · 2025-05-28
**Minimizing Pairwise Cross Entropy**

Great paper! I had a question about the ablative study portion. Specifically, the authors minimize pairwise cross entropy in order to determine the contribution of the consistency portion of the learning signal. Since cross entropy can be expressed as $H(p) + D_{KL}(p||q)$, wouldn't minimizing the cross entropy also minimize $H(p)$? Would you need to apply a stop gradient to the target distribution, $p$, in order to only enforce the consistency portion of the signal?

---

### Meta-Review · Area_Chair_QHv7 · 2022-08-28

**Recommendation:** Accept
**Confidence:** Certain

**Metareview:**

The authors propose MEMO, a heuristic for adapting classifiers at inference time based on a single test point. The adaptation procedure consists of constructing augmented versions of the test point and optimizing an objective that encourages "confidence and consensus" among the predictions on the augmented versions of the test point. The reviewers found the idea to be interesting, the experiments compelling, and were satisfied with the author responses during the discussion period. Thus, I am recommending acceptance.

**Award:**

No

---

### Decision · Program_Chairs · 2022-09-14

Accept